# The Role of Baseline Total Kidney Volume Growth Rate in Predicting Tolvaptan Efficacy for ADPKD Patients: A Feasibility Study

**DOI:** 10.3390/jcm14051449

**Published:** 2025-02-21

**Authors:** Hreedi Dev, Zhongxiu Hu, Jon D. Blumenfeld, Arman Sharbatdaran, Yelynn Kim, Chenglin Zhu, Daniil Shimonov, James M. Chevalier, Stephanie Donahue, Alan Wu, Arindam RoyChoudhury, Xinzi He, Martin R. Prince

**Affiliations:** 1Department of Radiology, Weill Cornell Medicine, New York, NY 10021, USA; hrd2001@med.cornell.edu (H.D.); zsh4001@med.cornell.edu (Z.H.); ars4017@med.cornell.edu (A.S.); yek4004@med.cornell.edu (Y.K.); chz4009@med.cornell.edu (C.Z.); xih4004@med.cornell.edu (X.H.); 2Rogosin Institute, New York, NY 10021, USA; jdblume@nyp.org (J.D.B.); das2041@nyp.org (D.S.); jac9014@nyp.org (J.M.C.); sld9001@nyp.org (S.D.); 3Department of Medicine, Weill Cornell Medicine, New York, NY 10021, USA; 4Division of Biostatistics, Department of Population Health, Weill Cornell Medicine, New York, NY 10021, USA; alw4001@med.cornell.edu (A.W.); arr2014@med.cornell.edu (A.R.); 5Department of Radiology, Columbia Vagelos College of Physicians and Surgeons, New York, NY 10032, USA

**Keywords:** ADPKD, tolvaptan, MRI, urine osmolality, total kidney volume, vasopressin, Mayo imaging classification, artificial intelligence, deep learning, treatment response

## Abstract

**Background/Objectives**: Although tolvaptan efficacy in ADPKD has been demonstrated in randomized clinical trials, there is no definitive method for assessing its efficacy in the individual patient in the clinical setting. In this exploratory feasibility study, we report a method to quantify the change in total kidney volume (TKV) growth rate to retrospectively evaluate tolvaptan efficacy for individual patients. Treatment-related changes in estimated glomerular filtration rate (eGFR) are also assessed. **Methods:** MRI scans covering at least 1 year prior to and during treatment with tolvaptan were performed, with deep learning facilitated kidney segmentation and fitting multiple imaging timepoints to exponential growth in 32 ADPKD patients. Clustering analysis differentiated tolvaptan treatment “responders” and “non-responders” based upon the magnitude of change in TKV growth rate. Differences in rate of eGFR decline, urine osmolality, and other parameters were compared between responders and non-responders. **Results:** Eighteen (56%) tolvaptan responders (mean age 42 ± 8 years) were identified by k-means clustering, with an absolute reduction in annual TKV growth rate of >2% (mean = −5.1% ± 2.5% per year). Thirteen (44%) non-responders were identified, with <1% absolute reduction in annual TKV growth rate (mean = +2.4% ± 2.7% per year) during tolvaptan treatment. Compared to non-responders, tolvaptan responders had significantly higher mean TKV growth rates prior to tolvaptan treatment (7.1% ± 3.6% per year vs. 3.7% ± 2.4% per year; *p* = 0.003) and higher median pretreatment spot urine osmolality (Uosm, 393 mOsm/kg vs. 194 mOsm/kg, *p* = 0.03), confirmed by multivariate analysis. Mean annual rate of eGFR decline was less in responders than in non-responders (−0.25 ± 0.04, CI: [−0.27, −0.23] mL/min/1.73 m^2^ per year vs. −0.40 ± 0.06, CI: [−0.43, −0.37] mL/min/1.73 m^2^ per year, *p* = 0.036). **Conclusions**: In this feasibility study designed to assess predictors of tolvaptan treatment efficacy in individual patients with ADPKD, we found that high pretreatment levels of annual TKV growth rate and higher pretreatment spot urine osmolality were associated with a responder phenotype.

## 1. Introduction

Tolvaptan, a selective type 2 vasopressin receptor inhibitor, was FDA-approved in 2018 for treatment of autosomal dominant polycystic kidney disease (ADPKD) patients at risk for rapidly progressive disease, based upon randomized clinical trials demonstrating that it reduced the annual rates of total kidney volume (TKV) growth and attenuated the rate of decline in estimated glomerular filtration rate (eGFR) [1,2,3,4,5,6,7]. In those clinical trials, TKV was estimated either by stereology or an ellipsoidal solid approximation with 5% to 18% measurement variability, respectively, making it difficult to evaluate the efficacy of tolvaptan on TKV growth rate in individual subjects [8,9]. Subsequent advances in deep learning techniques have enabled more accurate quantification of TKV, with measurement variability as low as 1.3% [10,11,12,13,14,15,16]. The assessment of TKV growth rate, based on multiple TKV measurements, has also become more precise due to the development of machine learning methodologies, which are more accurate than single measurement-based methods used in previous clinical trials [1,17,18,19,20]. Consequently, the efficacy of tolvaptan in individual patients with ADPKD can be more reliably quantified by comparing TKV growth rates prior to and during treatment.

In TEMPO 3:4, a randomized, placebo-controlled trial in which the efficacy of tolvaptan in treating ADPKD was first established, a parallel design was used [3]. However, TEMPO 3:4 did not determine factors that predicted a favorable therapeutic response in individual subjects. At that time, assessing tolvaptan efficacy for individual subjects was not practical due to the limitations in the measurements of TKV and estimation of eGFR. Ideally, multiple TKV measurements prior to and during tolvaptan treatment are required to establish whether a response to tolvaptan has occurred.

Tolvaptan treatment is expensive (about EUR 40,000 per year [21]), and has significant renal (e.g., thirst, polyuria) and extrarenal (e.g., liver toxicity) side effects that require controlled access and mandatory regularly scheduled liver function tests. Therefore, it would be useful to identify the ADPKD patients who are most likely to benefit from this treatment and to confirm treatment benefit after initiating tolvaptan therapy, thereby personalizing its use to maximize its benefit and limit harm.

Machine learning techniques improve the precision and reproducibility of MRI assessment of TKV and allow accurate calculation of TKV growth rates in ADPKD patients [17,18]. Furthermore, the deep learning methods are semi-automatic, reducing subjectivity and reducing the effort required for analyzing multiple studies. We hypothesize that machine learning measurements of TKV to calculate and compare baseline TKV growth rates with during tolvaptan TKV growth rates will identify individual patients responding to tolvaptan treatment.

In this study we evaluated TKV growth rates, and other aspects of the ADPKD phenotype, prior to and during treatment with tolvaptan to identify predictors of its efficacy in individual patients.

## 2. Materials and Methods

This retrospective review of existing data from ADPKD subjects prior to and during tolvaptan treatment is HIPAA-compliant and approved by the Weill Medical College IRB. The requirement for written informed consent was waived.

Inclusion criteria were: (1) ADPKD patients with symmetrical cystic kidney disease and at least 2 abdominal MRI scans, separated by at least 1 year, before starting tolvaptan treatment, to establish the baseline TKV growth rate and (2) at least two abdominal MRI scans performed after initiating tolvaptan, separated by at least 1 year, to establish the kidney growth rate during treatment. Exclusion criteria were: (1) any interruption in tolvaptan treatment of more than 1 month, (2) non-adherence with tolvaptan management, and (3) events that affect kidney growth such as cyst rupture/aspiration.

### 2.1. Clinical Data Collection

Patient demographic, laboratory and clinical data were extracted from the electronic medical record. Laboratory measurements were collected within one month prior to tolvaptan initiation, including serum creatinine, liver function tests (alanine aminotransferase (ALT), aspartate aminotransferase (AST)), and spot urine osmolality (Uosm). The 2021 CKD-EPI equation was used to calculate eGFR [22].

Patient encounter notes were reviewed to identify systolic and diastolic blood pressure, hypertension medication, and tolvaptan dose.

### 2.2. Response Status Classification

#### 2.2.1. Total Kidney Volume and Kidney Growth Rate Calculation

Left and right kidneys were contoured semi-automatically on abdominal MR scans by one expert (HD) for all scans as described previously using a 3D nnU-net deep learning model (www.traceorg.com). TKV was then computed as the sum of right kidney volume and left kidney volume. TKVs were calculated from all available pulse sequences that included both kidneys in their entirety, and the mean TKV across all available sequences, together with the age at the time of the scan were used to calculate kidney growth rates prior to and during treatment using the 2-parameter least squares fitting [18]. See Appendix A for kidney trajectories of each subject.

#### 2.2.2. Interobserver Variability

Three expert observers (H.D., Z.H., Y.K.) independently contoured kidneys and calculated kidney volumes of all available scans in a subgroup of 9 (28%) subjects. TKV growth rates were then calculated (as per Section 2.2.1) using the kidney volumes measured by each expert observer. Interobserver variability in kidney volume was assessed by calculating interclass correlation coefficients.

#### 2.2.3. Clustering Analysis

ADPKD subjects were designated as tolvaptan “responders” and “non-responders” based on the differences in TKV growth rates prior to, and during tolvaptan treatment. K-means clustering analysis was used to identify a natural separation between responders (in whom a larger magnitude of absolute decrease in the TKV growth rate occurred during treatment) and non-responders (in whom minimal or no decrease in the TKV growth rate occurred during treatment).

### 2.3. Estimated Kidney Function Trajectory

Serum creatinine was measured at monthly intervals for the first 18 months and then quarterly as routine monitoring of all patients receiving tolvaptan treatment, per standard care at our institution. eGFR was assessed over time for all subjects to determine whether the responder status for TKV growth rate corresponded to the rate of eGFR decline.

### 2.4. Statistics

Central tendency for continuous variables was calculated as mean ± standard deviation if they were normally distributed, as determined by the Shapiro–Wilk test. For non-normally distributed variables, median and interquartile range (IQR 25%/75%) are reported.

Student’s *t*-test was used to assess the statistical significance of differences between responders and non-responders for normally distributed continuous variables and the Mann–Whitney U/Wilcoxon rank sum test was used for the continuous variables that were not normally distributed. The Chi-squared test was used to assess the statistical significance of differences between responders and non-responders in the distribution of gender and Mayo Imaging Classification. Benjamini–Hochberg corrections was applied to *p*-values in the univariate analysis. Multivariate logistic regression models were constructed using the variables identified in the univariate analysis with uncorrected *p* < 0.05 together with age and gender to assess their association with responder and non-responder status where slope, standard error, and confidence interval of the slope, and the Variable Inflation Factor was calculated for each variable. The final model was informed based on a hybrid approach and the relative values of Akaike Information Criterion.

Based on TEMPO 3:4 results [3], we expected a 2.7% absolute reduction in annual TKV growth rate from a mean of 5.5% to 2.8% with 3% variability. Using an effective size = 0.9 as calculated by Cohen’s d for paired data, 90% power, alpha = 0.01, an estimated sample size of 18 subjects was required for this study to have adequate statistical power.

## 3. Results

Figure 1 indicates, 32 of 172 subjects initiated on tolvaptan fulfilled inclusion and exclusion criteria, with a sufficient number of MRI scans prior to and during tolvaptan treatment to determine the tolvaptan effect on TKV growth rate. Demographic and laboratory data are shown in Table 1, and a comparison to subjects not meeting study criteria show they have similar characteristics, Appendix A. Lower maximum tolvaptan median morning and evening doses reflect the fact that many of the excluded patients had stopped tolvaptan therapy before titrating up the maximum dose. The mean creatinine follow-up time on tolvaptan was 4 years and the mean imaging follow-up time was 3 years.

### 3.1. Interobserver Variability

For 28% of cases with kidney labels corrected independently by three expert observers, there was excellent label agreement with ICC = 0.998.

### 3.2. Majority of Subjects Responded to Tolvaptan

K-means clustering analysis identified one group (n = 14) with an absolute TKV growth rate change of <1% and another group (n = 18) with an absolute TKV growth rate change >2% (Figure 2). Accordingly, all 14 subjects with <1% absolute reduction in annual absolute TKV growth rate during treatment were designated “non-responders”. Eighteen exhibiting an absolute decrease in TKV growth rate >2% and were classified as “responders”. Demographic and laboratory characteristics of subjects grouped by responder status are shown in Table 1.

The mean annual TKV growth rate of all 32 subjects before tolvaptan was 5.7% ± 3.5% per year, decreasing to 3.8% ± 3.9% per year during tolvaptan treatment, as shown in Figure 3A, indicating that the cohort of 32 patients had a significant therapeutic response to tolvaptan therapy (*p* = 0.03). However, when the cohort was dichotomized as responders (Figure 3B) and non-responders (Figure 3C), the results diverged. The 18 responders had a mean pre-treatment annualized TKV growth rate of 7.1% ± 3.6%, which decreased to 2.0% ± 4.0% (*p* < 1 × 10^−6^) during tolvaptan treatment. In contrast, the 14 non-responders showed a mean pre-treatment annualized TKV growth rate of 3.7% ± 2.4%, which increased to 6.1% ± 2.3% (*p* = 0.006) during tolvaptan treatment. Examples of a responder and a non-responder are shown in Figure 4A,B, respectively.

### 3.3. Features of Tolvaptan Responders vs. Non-Responders

The clinical, laboratory and imaging features at the start of tolvaptan treatment are shown in Table 1. Responders are associated with a higher baseline TKV growth rate (7.1% ± 3.6% per year, *p* = 0.003) before starting tolvaptan than non-responders (3.7% ± 2.4% per year). Baseline spot Uosm was higher in responders (393 [256, 519] mOsm/kg, *p* = 0.03) compared to non-responders (194 [167, 239] mOsm/kg).

On multivariate analysis, baseline spot Uosm and baseline TKV growth rate were both associated with response status to tolvaptan (Table 2 and Appendix A).

There were no significant differences in the dosing of tolvaptan between responders and non-responders; 13 of 18 (72%) Responders and 13 of 14 (93%) non-responders were on the highest daily tolvaptan dose (90 mg AM/30 mg PM).

### 3.4. Responders Showed Slower Decline in eGFR than Non-Responders

During the first 4 years of tolvaptan treatment, eGFR, declined less rapidly in responders than in non-responders (−0.25 ± 0.04, CI: [−0.27, −0.23] mL/min/1.73 m^2^ per year vs. −0.40 ± 0.06, CI: [−0.43, −0.37] mL/min/1.73 m^2^ per year, *p* = 0.036; Figure 5).

## 4. Discussion

TEMPO 3:4 and REPRISE trials demonstrated that tolvaptan, a Type 2 vasopressin receptor antagonist, slows both TKV growth rate and the rate of eGFR decline in patients at high risk of rapidly progressing ADPKD [1,2,3,4,5,6,7]. However, these clinical trials did not differentiate the individual subject’s responses to tolvaptan. Moreover, sufficiently accurate and reproducible methods for measuring TKV, which is a biomarker of ADPKD progression, and for calculating TKV growth rate based on multiple imaging timepoints before and during tolvaptan within individual subjects were not available when those Phase 3 studies were performed. In the current feasibility study of 32 patients at high risk of rapidly progressing ADPKD, measurements of TKV growth rate were calculated from multiple timepoints using a more reproducible deep learning-based MRI TKV measurement technique [17,18]. We found that tolvaptan reduced the annual TKV growth rate in 56% of subjects (responders) and failed to slow the TKV growth rate in 44% (non-responders). This change in TKV growth rate response corresponded with the change in the eGFR response; those patients with a decrease in TKV growth rate during tolvaptan treatment also had a decrease in the rate of eGFR decline. Conversely, tolvaptan non-responders had a more rapid decline in eGFR compared to responders.

Responders were indistinguishable from non-responders by their baseline Mayo Clinic Imaging Classification (MCIC), a predictor of rate of progressive kidney disease in ADPKD that is based on height-adjusted kidney volume and patient age [1]. However, in our study, responders were associated with higher TKV growth rates prior to tolvaptan treatment compared to non-responders. Therefore, our data suggest that the MCIC is an inadequate marker for predicting therapeutic efficacy of tolvaptan in ADPKD. Accordingly, we propose that establishing a baseline TKV growth rate prior to treatment by using multiple imaging timepoints would appear to be a useful strategy for predicting the potential benefit of tolvaptan in individual ADPKD patients. Similarly, annual MRI scanning during tolvaptan with TKV measurements by a highly reproducible measurement technique appears to confirm whether a treatment response has occurred. If confirmed by prospective studies, this would have major prognostic implications, as the TKV growth rate response to tolvaptan corresponds with the trajectory of the eGFR response to tolvaptan treatment [3,4,5].

High Uosm is indicative of a high vasopressin level and has been identified as a potential biomarker for the long-term efficacy of tolvaptan in ADPKD by predicting the effects on TKV and eGFR [23]. A post hoc analysis of the TEMPO 3:4 trial reported that short-term reductions in Uosm correlated with slower rates of decline in kidney function, similar to the responders in our study [24]. Moreover, we recently showed in ADPKD patients that high water intake, sufficient to decrease mean 24 h Uosm and serum copeptin levels [25], decreased the height-adjusted TKV annual growth rate [26]. Vasopressin (via cyclic AMP) stimulates both urinary concentrating ability and kidney cyst growth and most likely accounts for the significant association of pretreatment TKV growth rate with the decrease in TKV growth rate during tolvaptan treatment. Spot Uosm measurements are variable and dependent upon the level of hydration and other factors at the time of sample collection. The broad confidence interval for the effect of baseline spot Uosm on determination of tolvaptan response status suggests further investigation of this effect in larger studies is needed, preferably measuring 24 h Uosm, which is a more stable indicator of chronic vasopressin activity than a spot Uosm. By contrast, TKV is a stable and reproducible measurement at any time point and is not affected acutely by transient factors such as hydration and solute intake.

Although urine specific gravity is sometimes used in place of Uosm because of their correlation, in our study, the baseline urine specific gravity showed only a very weak trend toward being higher in responders compared to non-responders (*p* = 0.22). This may reflect the fact that urine specific gravity is affected by the type of solute present, whereas Uosm is not affected by the type of solute. Thus, urine specific gravity should not be used in place of baseline Uosm for assessing the likelihood of response to tolvaptan.

Several other factors may contribute to the heterogeneity in responsiveness to tolvaptan. Although non-adherence to treatment is a consideration, it is unlikely that these subjects would be adherent to extensive imaging procedures and laboratory testing, but not to their treatment. Another consideration to account for the tolvaptan response phenotype is that kidney cysts originate from renal tubule epithelium from all nephron segments and are heterogeneous in size, appearance, and cellular characteristics. Although cortical collecting duct cysts, which are responsive to vasopressin, are predominant in ADPKD, cysts also originate from other nephron segments that are not responsive to vasopressin [27]. Therefore, it is conceivable that tolvaptan efficacy is related to the prevalence of vasopressin-responsive cysts in the individual patient.

We included patient age and sex in our multivariate analysis of tolvaptan response predictors because these factors are traditionally considered in assessing ADPKD progression [28]. However, neither age nor sex was a significant predictor of tolvaptan response in either univariate or multivariate analyses. This emphasizes that objective biomarkers such as baseline TKV growth rate and Uosm may be more relevant for predicting treatment efficacy than demographic characteristics.

The strengths of this study include: (i) multiple MRI scan data available both prior to and during tolvaptan treatment, (ii) MRI TKV measurement techniques that employed machine learning methodology that has been validated previously in large cohorts of ADPKD patients by experienced observers, and (iii) long periods of follow-up prior to and during tolvaptan treatment.

Important limitations are the relatively small sample size and retrospective data collection. This limited our study to reviewing MRI scans, the main imaging technique used at our institution to monitor ADPKD patients, because there were no subjects with sufficient CT scans to measure TKV growth rate both prior to and during tolvaptan. Ultrasound could not be used to track kidney volume because the variability of TKV measurement by the ultrasound equipment available at our center was too large to obtain meaningful TKV growth rate information. Another limitation is the lack of cross-validation or external/independent validation. Also, there was no adjustment for potential confounders in the assessment of eGFR decline, such as blood pressure, glycemic control, or concomitant medications. Although we were unable to use laboratory testing to assess adherence with their tolvaptan prescription, all patients were compliant with monthly clinically indicated laboratory testing and regular follow-up appointments with their nephrologist, which were required to maintain ongoing tolvaptan treatment. Patients fulfilling the inclusion/exclusion criteria (32 out of 172, 18%) were on a higher median tolvaptan dose compared to excluded patients on tolvaptan. This could have caused an increase in the proportion of responders, if patients on higher doses have a better treatment response, and is recommended to be explored in future studies.

## 5. Conclusions

Careful measurements of total kidney volume, using a highly reproducible MRI-based technique and evaluation of TKV growth rates prior to and during tolvaptan treatment by fitting TKV measurements to multiple timepoints, showed that more than 40% of ADPKD subjects did not significantly decrease their annualized TKV growth rate, a biomarker of disease severity and prognosis, during a 4-year period of treatment with tolvaptan. The tolvaptan responder group, defined by a higher mean pretreatment, baseline TKV growth rate than non-responders, also had a slower rate of decline in estimated glomerular filtration rate during tolvaptan treatment. Thus, baseline TKV growth rate appeared to predict tolvaptan efficacy in individual patients, and the difference in TKV growth rate prior to and during tolvaptan can be used to evaluate tolvaptan efficacy in individuals. The small sample size, retrospective study design, and lack of external validation may limit generalization of the results. Prospective studies of larger populations of ADPKD patients are required to confirm these findings. Genetic and molecular biomarkers may complement baseline TKV growth rate and baseline Uosm for predicting tolvaptan treatment response.

## Figures and Tables

**Figure 1 jcm-14-01449-f001:**
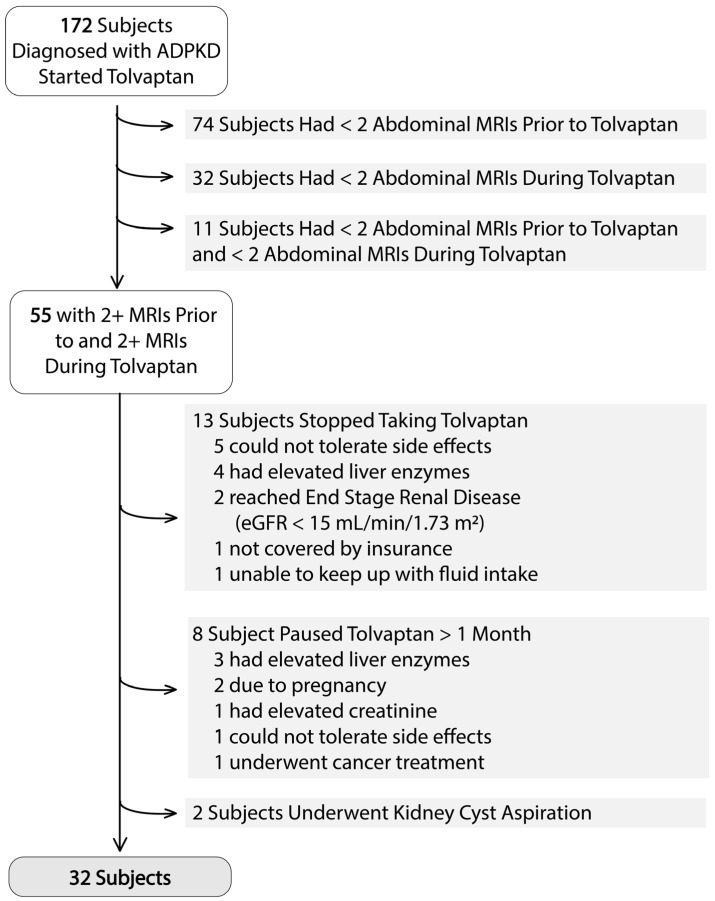
Patient flowchart.

**Figure 2 jcm-14-01449-f002:**
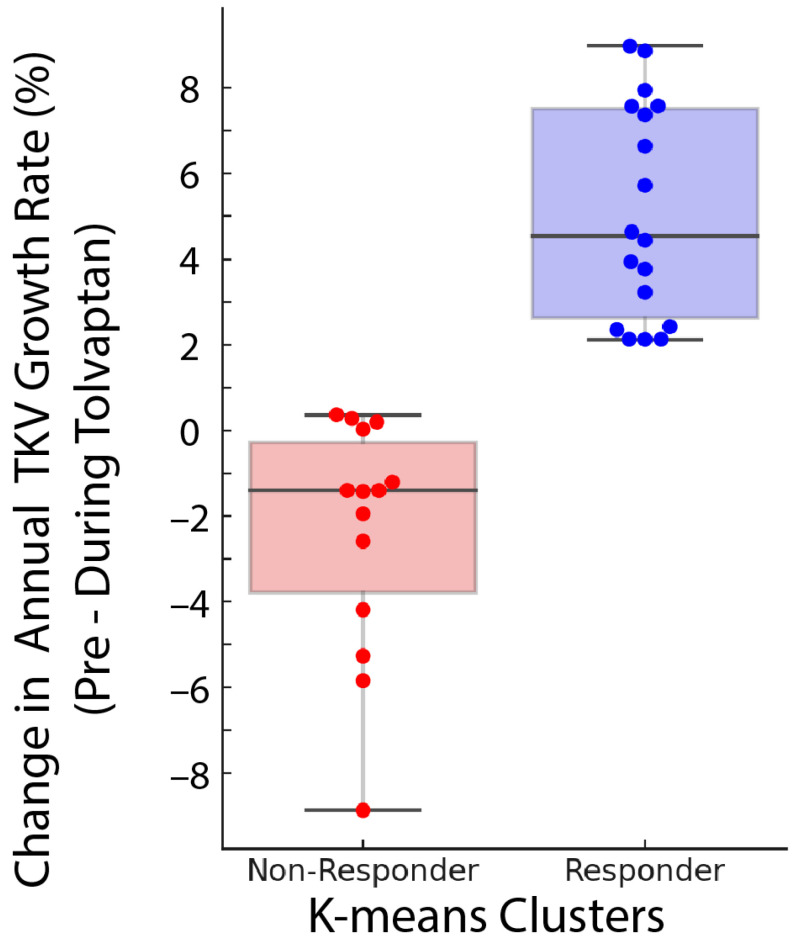
Clustering analysis of absolute change in TKV growth rate before and during tolvaptan treatment.

**Figure 3 jcm-14-01449-f003:**
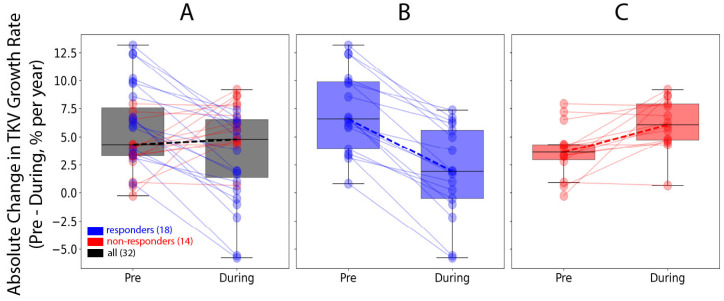
Change in TKV growth rate from pre to during tolvaptan for the entire cohort ((**A**), grey), responders ((**B**), blue), and non-responders ((**C**), red). Note that all responders (n = 18) had at least 2% absolute reduction in the TKV growth rate while non-responders (n = 14) all had less than 1% absolute reduction in TKV growth rate.

**Figure 4 jcm-14-01449-f004:**
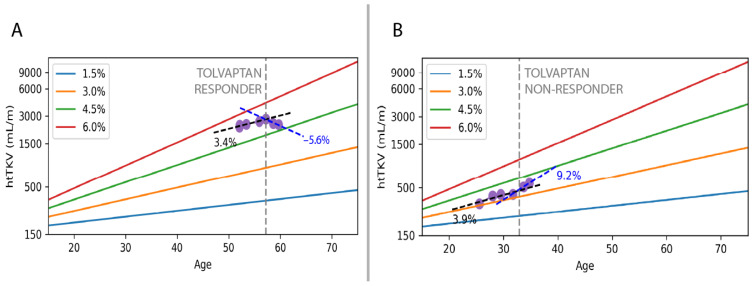
Change in TKV growth rate during tolvaptan treatment in a responder ((**A**), 3.4% to −5.6% TKV absolute change per year) and non-responder ((**B**), 3.9% to 9.2% TKV absolute change per year). Time of tolvaptan initiation is indicated by the vertical grey dashed line. Pre-tolvaptan TKV growth rate (black dashed line) and during-tolvaptan TKV growth rate (blue dashed line) were calculated using two-parameter least squares fitting of TKV measured on MRIs acquired before and after tolvaptan initiation respectively. Blue, orange, green and red lines correspond to Mayo Imaging Classification trajectories with annual TKV growth rates in the legend.

**Figure 5 jcm-14-01449-f005:**
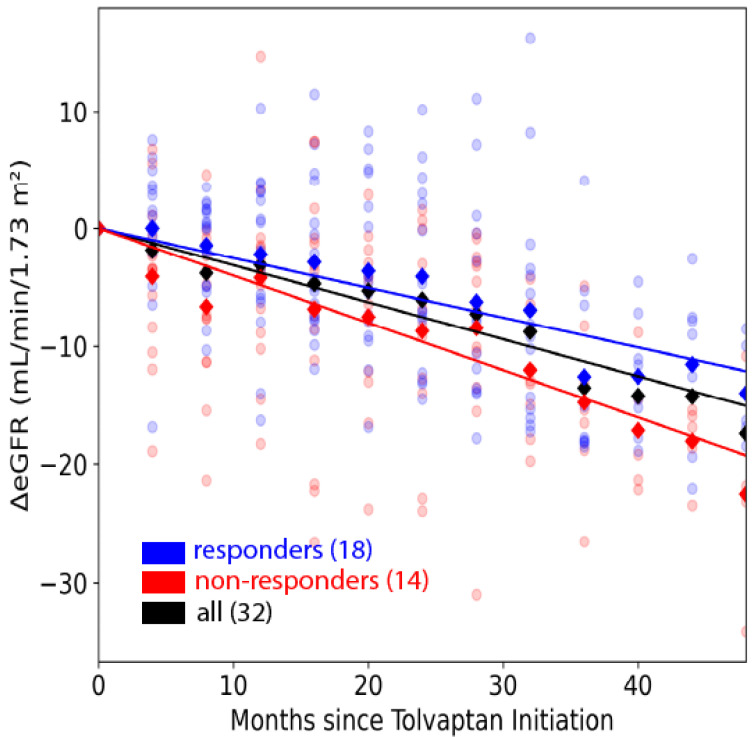
Trajectory of estimated glomerular filtration rate (eGFR) in 32 subjects. Change in eGFR (ΔeGFR) measurements of responders (blue dots) and non-responders (red dots) every 4 month since tolvaptan initiation to month 48 overlayed by fitted trendlines for all patients (black line), responders (blue line), non-responders (red line). Rate of eGFR decline in responders is −0.25 ± 0.04, CI: [−0.27, −0.23] mL/min/1.73 m^2^ per year, and rate of eGFR decline in non-responders is −0.40 ± 0.06, CI: [−0.43, −0.37] mL/min/1.73 m^2^ per year, which is 60% greater than responders.

**Table 1 jcm-14-01449-t001:** Baseline demographic and laboratory data.

	All Subjects (*n* = 32)	Responder (*n* = 18)	Non-Responder (*n* = 14)	*p*-Value * Responder vs. Non-Responder
Age (years)	42 ± 9	42 ± 8	42 ± 11	0.93
Sex (male:female, %male)	17:15, 53%	10:8, 56%	7:7, 50%	1.00
Height (m)	1.7 ± 0.1	1.7 ± 0.1	1.7 ± 0.1	0.59
Weight (kg)	75 ± 16	76 ± 19	73 ± 13	0.61
BMI ^1^ (kg/m^2^)	26 ± 4	26 ± 5	25 ± 3	0.51
TKV (mL)	1622 [1067, 2715]	1616 [918, 2569]	1819 [1172, 2698]	0.69
htTKV (mL/m)	975 [589, 1578]	954 [538, 1519]	1019 [700, 1596]	0.66
Mayo Imaging Classification 1A 1B 1C 1D 1E	0 (0%) 2 (6%) 16 (50%) 8 (25%) 6 (19%)	0 (0%) 2 (11%) 9 (50%) 3 (17%) 4 (22%)	0 (0%) 0 (0%) 7 (50%) 5 (36%) 2 (14%)	0.58
Blood Pressure (mmHg) ^2^ Systolic Diastolic	126 ± 13 77 ± 8	127 ± 14 77 ± 7	124 ± 12 78 ± 8	0.51 0.86
Liver Function Tests AST ^3^ (U/L) ALT ^4^ (U/L) Bilirubin Total (mg/dL)	22 [19, 27] 22 [18, 27] 0.6 [0.5, 0.7]	22 [18, 28] 23 [20, 28] 0.6 [0.6, 0.7]	22 [21, 25] 20 [18, 26] 0.6 [0.5, 0.7]	0.82 0.45 0.75
Serum Creatinine (mg/dL)	1.3 ± 0.5	1.3 ± 0.4	1.3 ± 0.5	0.83
eGFR ^5^ (mL/min/1.73 m^2^)	67 ± 27	68 ± 26	67 ± 29	0.96
Urine Specific Gravity	1.009 [1.005, 1.012]	1.010 [1.008, 1.013]	1.006 [1.005, 1.009]	0.22
Spot Uosm (mOsm/kg)	256 [168, 460]	393 [256, 519]	194 [167, 239]	0.03
Tolvaptan Morning Dose	90 [90, 90]	90 [68, 90]	90 [90, 90]	0.19
Tolvaptan Evening Dose	30 [30, 30]	30 [30, 30]	30 [30, 30]	0.73
Pre-tolvaptan TKV Growth Rate (% per year)	5.7 ± 3.5	7.1 ± 3.6	3.7 ± 2.4	0.003

^1^ Body mass index; ^2^ Blood pressure was not recorded in four subjects at tolvaptan initiation; ^3^ Aspartate aminotransferase; ^4^ Alanine transaminase; ^5^ Estimated glomerular filtration rate; * None of these uncorrected *p*-values were significant after Benjamini–Hochberg corrections.

**Table 2 jcm-14-01449-t002:** Multivariate logistic regression assessment of parameters associated with tolvaptan responder status. This model is selected based on a hybrid process, where all multivariate models assessed can be found in Appendix A.

Variable	Coefficient	Standard Error	95% Confidence Interval	*p*-Value	VIF ^1^
(Intercept)	4.2	1.6	[1.06, 7.3]	0.01 *	-
Baseline TKV Growth Rate	−0.43	0.19	[−0.81, −0.06]	0.02 *	1.14
Baseline Spot Uosm	−0.006	0.004	[0.014, −0.001]	0.03 *	1.14

^1^ Variable Inflation Factor; * *p* ≤ 0.05.

## Data Availability

Raw data can be shared under a data sharing agreement negotiated with Weill Cornell Medicine.

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
