# Peer review of "The Role of Baseline Total Kidney Volume Growth Rate in Predicting Tolvaptan Efficacy for ADPKD Patients: A Feasibility Study"

_jcm, 2025, doi:10.3390/jcm14051449_

Round 1

Reviewer 1 Report

Comments and Suggestions for Authors

To the authors,

Thank you for the opportunity to review manuscript. This study presents a promising analysis of the characteristics of patients with ADPKD who responded to tolvaptan treatment, which has significant implications for clinical practice. However, I propose the following revisions before the manuscript can be accepted.

<Specific comments>

1. This study excluded many patients because of the use of MRI to measure the kidney volume. Therefore, it is necessary to clarify the rationale for selecting MRI over CT or ultrasound. If MRI is performed only on patients with specific conditions, the generalizability of the results to a broader patient population may be limited.

2. While it is reasonable that baseline urine osmolality is associated with tolvaptan responsiveness, why is urine specific gravity not related? A discussion addressing this discrepancy will strengthen our manuscript.

Author Response

Thank you for the opportunity to review manuscript. This study presents a promising analysis of the characteristics of patients with ADPKD who responded to tolvaptan treatment, which has significant implications for clinical practice. The authors thank reviewer 1 for this positive comment.

However, I propose the following revisions before the manuscript can be accepted. 

  1. This study excluded many patients because of the use of MRI to measure the kidney volume. Therefore, it is necessary to clarify the rationale for selecting MRI over CT or ultrasound. If MRI is performed only on patients with specific conditions, the generalizability of the results to a broader patient population may be limited. We now clarify in the discussion (7th paragraph) that at our institution, MRI is the main imaging technique used for tracking TKV. If we were to use CT, then no patients would be eligible for the analysis. If we were to use ultrasound, the measurement variability would be too high to accurately measure TKV growth rates.

  1. While it is reasonable that baseline urine osmolality is associated with tolvaptan responsiveness, why is urine specific gravity not related? A discussion addressing this discrepancy will strengthen our manuscript. We have added 4 additional cases since the previous submission and the data now shows a weak trend toward higher urine specific gravity in responders, 1.010 compared to a lower specific gravity in non-responders, 1.006 (p = 0.22). We now discuss why specific gravity is not identical to urine osmolality and that it should not substitute for Uosm in assessing for the likelihood of response to tolvaptan.

Reviewer 2 Report

Comments and Suggestions for Authors

Dear Authors,

Respectfully, I have shared my comments and suggestions, which I hope will contribute to enhancing the clarity, coherence, and scientific rigor of your manuscript. Below, I outline key areas for improvement and propose practical recommendations.

Title

Comment: The current title reflects the essence of the study but can be refined to convey greater clarity, scientific precision, and highlight the key findings of the work. For example: “The Role of Baseline Total Kidney Volume Growth Rate in Predicting Tolvaptan Efficacy for ADPKD Patients: A Feasibility Study.”

Introduction

Comment: Although the study mentions the difficulty of identifying patients who would benefit most from tolvaptan, it does not clearly present the central problem. The absence of reliable methods to predict treatment response represents a significant obstacle in clinical practice, especially considering the high cost and adverse effects of the drug.

Comment: The introduction acknowledges the high cost and adverse effects of tolvaptan but does not explore the clinical and economic impact in detail. Highlighting the average cost of treatment, challenges in adherence due to side effects, and the importance of personalizing the use of the drug would help reinforce the practical relevance of the study.

Comment: Despite citing limitations of studies such as TEMPO 3:4, it is unclear what specific gaps this study aims to address. TEMPO 3:4 focused on population-level outcomes but did not investigate individual predictors. Explaining how this work contributes by exploring markers such as baseline TKV and eGFR would strengthen its justification.

Comment: The introduction mentions the identification of patients most likely to respond to treatment but does not explain how the retrospective study design aligns with this objective. Why was a retrospective study chosen instead of a prospective one? How do recent technologies, such as machine learning, justify this retrospective approach?

Objective

Comment: The objective does not explicitly mention the retrospective design and the exploratory nature of the study.

2.4 Statistics

Comment: Out of 141 patients initially assessed, only 28 met the inclusion/exclusion criteria. Although the statistical methods employed are theoretically adequate, their application to such a small dataset without rigorous adjustments may compromise the robustness of the results. I suggest the authors revisit the statistical analysis to ensure greater confidence in the conclusions. This is especially important in a study with potential clinical impact like this one.

Comment: The lack of a statistical power calculation prevents a clear assessment of whether the study has sufficient data to support its conclusions.

Comment: Several univariate tests were performed (e.g., differences in baseline characteristics, analysis by gender, and imaging classification). The absence of adjustments for multiple comparisons (e.g., Benjamini-Hochberg) increases the risk of type I errors (false positives).

Comment: The approach of including variables with p < 0.05 in the univariate analysis may be insufficient to handle collinearity among variables. There is no mention of using metrics such as the Variance Inflation Factor (VIF) to assess whether the model's variables are correlated, which could distort the coefficients.

Comment: Although coefficients were presented in the logistic regressions, confidence intervals for the coefficients were not provided. This makes it difficult to assess the precision and robustness of the results.

Comment: There was no mention of cross-validation or independent validation for the multivariate models. This is particularly important in studies with small samples to avoid overfitting.

Results

3.2. Majority of subjects responded to tolvaptan

Comment: The chosen cutoff points (>2% and <1% change in TKV growth rate) lack external validation. This reduces the applicability of the findings and confidence in the patient categorization.

Comment: Several univariate analyses were performed, increasing the risk of false positives (type I error). Adjustments such as Benjamini-Hochberg should have been applied.

Comment: Although mentioned in the study's objective, eGFR did not show significant differences between groups and appears to play a secondary role.

3.3. Features of Tolvaptan Responders vs. Non-Responders

Comment: Although significant in the univariate analysis, urinary osmolality lost relevance in the multivariate model, indicating that it is not a robust independent marker. In models excluding baseline TKV growth rate, osmolality becomes significant again, suggesting possible collinearity between these variables that was not adequately explored (e.g., using VIF - Variance Inflation Factor).

Comment: The multivariate table does not present confidence intervals for the coefficients, making it difficult to interpret the precision of the results.

Comment: The arbitrary classification of Responders (>2% reduction) and Non-Responders (<1% reduction) in TKV growth rate may influence the interpretation of the identified predictors. A validated definition would be more robust.

3.4. Responders showed slower decline in eGFR than Non-Responders

Comment: Responders showed a significantly slower decline in eGFR compared to Non-Responders, aligning with findings from studies such as TEMPO 3:4 and REPRISE. However, it is unclear whether the decline in eGFR was adjusted for potential confounding factors such as blood pressure, glycemic control, or concomitant medications (e.g., ACE inhibitors or angiotensin receptor blockers). These factors could influence eGFR progression independently of tolvaptan.

Comment: Reporting confidence intervals for eGFR decline rates would increase the transparency and reliability of the estimates.

Discussion

Comment: Although the discussion mentions that urinary osmolality was not significant in the multivariate model, it dedicates excessive space to explaining its relationship with vasopressin and its role in other studies.

Comment: While the study showed that 46% of patients did not respond to tolvaptan, the discussion does not adequately explore the possible reasons for this.

Comment: The discussion mentions that adherence was not directly assessed but relies on compliance with monthly laboratory tests as a substitute. This assumption is insufficient, considering that the efficacy of tolvaptan depends on rigorous adherence due to its complex administration and adverse effects.

Conclusion

Comment: It is necessary to include that despite these interesting findings, the small sample size, the retrospective nature of the study, and the lack of external validation limit the generalization of the results.

Comment: Recognize the need to explore other markers beyond TKV. Future investigations should consider genetic and molecular biomarkers that could complement baseline TKV as predictors of response to tolvaptan.

Kind regards.

Author Response

Respectfully, I have shared my comments and suggestions, which I hope will contribute to enhancing the clarity, coherence, and scientific rigor of your manuscript. Below, I outline key areas for improvement and propose practical recommendations.  Thank you.

Title

  1. Comment: The current title reflects the essence of the study but can be refined to convey greater clarity, scientific precision, and highlight the key findings of the work. For example: “The Role of Baseline Total Kidney Volume Growth Rate in Predicting Tolvaptan Efficacy for ADPKD Patients: A Feasibility Study.”

We have now adopted the title as you have proposed.

Introduction

  1. Comment: Although the study mentions the difficulty of identifying patients who would benefit most from tolvaptan, it does not clearly present the central problem. The absence of reliable methods to predict treatment response represents a significant obstacle in clinical practice, especially considering the high cost and adverse effects of the drug. This point is now made in the introduction, 2nd paragraph.

  1. Comment: The introduction acknowledges the high cost and adverse effects of tolvaptan but does not explore the clinical and economic impact in detail. Highlighting the average cost of treatment, challenges in adherence due to side effects, and the importance of personalizing the use of the drug would help reinforce the practical relevance of the study. The annual estimated cost of tolvaptan is now indicated in the introduction, 2nd paragraph (citing a recent 2025 paper, reference 22), along with the challenge of adherence due to side effects and the importance of personalizing use of the drug.

  1. Comment: Despite citing limitations of studies such as TEMPO 3:4, it is unclear what specific gaps this study aims to address. TEMPO 3:4 focused on population-level outcomes but did not investigate individual predictors. Explaining how this work contributes by exploring markers such as baseline TKV and eGFR would strengthen its justification. We now explain more clearly in the introduction, the gaps in the prior studies that are being addressed. Specifically, the TEMPO3:4 did not measure baseline TKV growth rate because at that time accurate, reproducible measurement of TKV growth rate was not practical with the available techniques.

  1. Comment: The introduction mentions the identification of patients most likely to respond to treatment but does not explain how the retrospective study design aligns with this objective. Why was a retrospective study chosen instead of a prospective one? How do recent technologies, such as machine learning, justify this retrospective approach? We now explain how the retrospective study design enables us to go back in time and explore how TKV growth rate calculated from multiple TKV measurements during as well as pre-dating initiation of tolvaptan treatment can help assess and predict treatment response.

Objective

  1. Comment: The objective does not explicitly mention the retrospective design and the exploratory nature of the study. The objective (beginning of abstract) now mentions the retrospective design and exploratory nature of the study.

2.4 Statistics

  1. Comment: Out of 141 patients initially assessed, only 28 met the inclusion/exclusion criteria. Although the statistical methods employed are theoretically adequate, their application to such a small dataset without rigorous adjustments may compromise the robustness of the results. I suggest the authors revisit the statistical analysis to ensure greater confidence in the conclusions. This is especially important in a study with potential clinical impact like this one. As more patients have undergone additional follow-up scans there are now 4 additional patients meeting the inclusion/exclusion criteria and they have been incorporated into the analysis. We now include supplemental table S1 which shows how the patients initiating tolvaptan therapy but not included (n = 140) in the study compared to the 32 finally meeting inclusion/exclusion criteria.  There was a small difference at ALT, 20 vs 21, but both were still in the normal range and there was a difference in tolvaptan dose reflecting the excluded group having more subjects who stopped tolvaptan while still on a low dose prior to titrating up to the highest dose. As these two groups are similar, no adjustments are necessary.

  1. Comment: The lack of a statistical power calculation prevents a clear assessment of whether the study has sufficient data to support its conclusions. We now provide the power calculation addressing the issue as to how many subjects are needed to have sufficient statistical power to show that reduction in TKV growth rate on tolvaptan is statistically significant. Based on TEMPO 3:4 results, we are expecting a 2.7% absolute reduction in annual TKV growth rate from a mean of 5.5% down to 2.8% with 3% variability. Using an effective size = 0.9 as calculated by Cohen’s d for paired data, 90% power, alpha = 0.01, 18 subjects are required.

  1. Comment: Several univariate tests were performed (e.g., differences in baseline characteristics, analysis by gender, and imaging classification). The absence of adjustments for multiple comparisons (e.g., Benjamini-Hochberg) increases the risk of type I errors (false positives). We now indicate that none of the univariate p values are significant after Benjamini-Hochberg correction (see Table 1 and 2.4 Statistics 2nd paragraph). As the univariate analysis is not the final results, we still use the uncorrected univariate p values to select parameters for inclusion in the multivariate analysis.

  1. Comment: The approach of including variables with p < 0.05 in the univariate analysis may be insufficient to handle collinearity among variables. There is no mention of using metrics such as the Variance Inflation Factor (VIF) to assess whether the model's variables are correlated, which could distort the coefficients. This was not used for the univariate analysis since that was not the final result. However, we have now added VIF to the Multivariate analysis, Table 2 and 2.4 Statistics, 2nd paragraph.

  1. Comment: Although coefficients were presented in the logistic regressions, confidence intervals for the coefficients were not provided. This makes it difficult to assess the precision and robustness of the results. Confidence intervals are now provided.

  1. Comment: There was no mention of cross-validation or independent validation for the multivariate models. This is particularly important in studies with small samples to avoid overfitting. This is an important point but beyond the scope of this manuscript and is now mentioned in the limitations section of the discussion.

Results

3.2. Majority of subjects responded to tolvaptan

  1. Comment: The chosen cutoff points (>2% and <1% change in TKV growth rate) lack external validation. This reduces the applicability of the findings and confidence in the patient categorization. This important point is now mentioned as a limitation in the discussion.
  2. Comment: Several univariate analyses were performed, increasing the risk of false positives (type I error). Adjustments such as Benjamini-Hochberg should have been applied. We now indicate that none of the univariate p values are significant after Benjamini-Hochberg correction (see Table 1 and 2.4 Statistics 2nd paragraph). As the univariate analysis is not the final results, we still use the uncorrected univariate p values to select parameters for inclusion in the multivariate analysis.
  3. Comment: Although mentioned in the study's objective, eGFR did not show significant differences between groups and appears to play a secondary role. With our new analysis including 4 additional subjects, the rate of eGFR decline is statistically significantly slower for responders compared to non-responders (p = 0.036), see section 3.4 and Figure 5.

3.3. Features of Tolvaptan Responders vs. Non-Responders

  1. Comment: Although significant in the univariate analysis, urinary osmolality lost relevance in the multivariate model, indicating that it is not a robust independent marker. In models excluding baseline TKV growth rate, osmolality becomes significant again, suggesting possible collinearity between these variables that was not adequately explored (e.g., using VIF - Variance Inflation Factor). We have now added VIF to the Multivariate analysis, Table 2 and 2.4 Statistics, 2nd paragraph. With the increase in sample size, baseline Uosm is now significant in both univariate and multivariate analyses.

  1. Comment: The multivariate table does not present confidence intervals for the coefficients, making it difficult to interpret the precision of the results. Confidence intervals are now provided.

  1. Comment: The arbitrary classification of Responders (>2% reduction) and Non-Responders (<1% reduction) in TKV growth rate may influence the interpretation of the identified predictors. A validated definition would be more robust. We agree with this important point, however no validated definition is currently available which is why this exploratory work is important. Section 2.2.3 (and also 3.2) clarifies that this classification is based upon k-means cluster analysis (not arbitrary).   

3.4. Responders showed slower decline in eGFR than Non-Responders

  1. Comment: Responders showed a significantly slower decline in eGFR compared to Non-Responders, aligning with findings from studies such as TEMPO 3:4 and REPRISE. However, it is unclear whether the decline in eGFR was adjusted for potential confounding factors such as blood pressure, glycemic control, or concomitant medications (e.g., ACE inhibitors or angiotensin receptor blockers). These factors could influence eGFR progression independently of tolvaptan. We now point out in the discussion (7th paragraph) that a limitation of this work is that the analysis of eGFR decline in responders vs. non-responders could not be adjusted for potential confounders such as blood pressure, glycemic control or concomitant medications.
  2. Comment: Reporting confidence intervals for eGFR decline rates would increase the transparency and reliability of the estimates. Confidence intervals are now reported in addition to the standard deviations included previously.

Discussion

  1. Comment: Although the discussion mentions that urinary osmolality was not significant in the multivariate model, it dedicates excessive space to explaining its relationship with vasopressin and its role in other studies. Now that 4 additional patients are added to the data set, baseline Uosm is significant in both univariate and multivariate analysis (previously the multivariate analysis was borderline). So now it makes sense to leave that discussion as is.

  1. Comment: While the study showed that 46% of patients did not respond to tolvaptan, the discussion does not adequately explore the possible reasons for this. We now further explore the possible reasons for lack of treatment response to tolvaptan in the discussion (5th paragraph).

  1. Comment: The discussion mentions that adherence was not directly assessed but relies on compliance with monthly laboratory tests as a substitute. This assumption is insufficient, considering that the efficacy of tolvaptan depends on rigorous adherence due to its complex administration and adverse effects. We agree that this is a limitation to the study, so it is now included in the discussion (7th paragraph).

Conclusion

  1. Comment: It is necessary to include that despite these interesting findings, the small sample size, the retrospective nature of the study, and the lack of external validation limit the generalization of the results.  This point is now added to the conclusion.
  1. Comment: Recognize the need to explore other markers beyond TKV. Future investigations should consider genetic and molecular biomarkers that could complement baseline TKV as predictors of response to tolvaptan.  This point is now added to the conclusion.

Round 2

Reviewer 1 Report

Comments and Suggestions for Authors

To the authors,

Thank you for the opportunity to review the revised manuscript and for taking into consideration my suggestions. This study was dramatically improved, and I find the revised version much more suitable for publication.

Author Response

Thank you for the opportunity to review the revised manuscript and for taking into consideration my suggestions. This study was dramatically improved, and I find the revised version much more suitable for publication.  

The authors thank reviewer 1 for this positive comment.

Reviewer 2 Report

Comments and Suggestions for Authors

Dear Authors,

Thank you for your responses to my comments and suggestions.

The article has significantly improved in terms of clarity, statistical robustness, and transparency regarding its limitations.

Congratulations on the balanced tone and the appropriate interpretation of the results.

My final suggestions:

Discussion Section (Lines 239-243): The study found that the included patients received significantly higher doses of tolvaptan than those excluded (p = 0.02 for the morning dose and p = 0.04 for the evening dose).

Suggestion: This factor was not discussed but is relevant, as higher doses could contribute to a better treatment response (as observed in Table S1). This may have influenced the proportion of responders, but the authors did not address it. I suggest including this point in the discussion and recommending that it be explored in future studies.

Discussion Section (Lines 251-255): The authors correctly focused on the importance of baseline TKV growth rate and Uosm as predictors of response to tolvaptan.

Suggestion: Age and gender are traditionally considered factors in the progression of Autosomal Dominant Polycystic Kidney Disease (ADPKD). However, in the present study, neither of these factors was a significant predictor of response to tolvaptan, as demonstrated in the multivariate analysis. Highlighting this finding in the discussion would reinforce that objective biomarker, such as baseline TKV growth rate and Uosm, may be more relevant for predicting treatment efficacy than demographic characteristics.

Discussion Section (Lines 262-265): The authors mention that urinary osmolality (Uosm) may predict the efficacy of tolvaptan, citing a post-hoc analysis of the TEMPO 3:4 study, which correlated reductions in Uosm with a slower decline in kidney function.

Suggestion: The authors correctly discuss the relevance of Uosm as a predictive biomarker. However, they could have explicitly linked this finding to Table S2, where Baseline Spot Uosm was identified as a significant predictor of response to tolvaptan (p = 0.03) in the logistic regression analysis. Additionally, the confidence interval for this effect (-0.014, -0.001) suggests that the relationship between this marker and treatment response still needs further investigation in larger studies.

Best regards.

Author Response

  1. Discussion Section (Lines 239-243): The study found that the included patients received significantly higher doses of tolvaptan than those excluded (p = 0.02 for the morning dose and p = 0.04 for the evening dose).
    Suggestion: This factor was not discussed but is relevant, as higher doses could contribute to a better treatment response (as observed in Table S1). This may have influenced the proportion of responders, but the authors did not address it. I suggest including this point in the discussion and recommending that it be explored in future studies. Thank you for this important point which is now included in the discussion (see discussion, paragraph 8) along with a recommendation to be explored in future studies.
  2. Discussion Section (Lines 251-255): The authors correctly focused on the importance of baseline TKV growth rate and Uosm as predictors of response to tolvaptan.
    Suggestion: Age and gender are traditionally considered factors in the progression of Autosomal Dominant Polycystic Kidney Disease (ADPKD). However, in the present study, neither of these factors was a significant predictor of response to tolvaptan, as demonstrated in the multivariate analysis. Highlighting this finding in the discussion would reinforce that objective biomarker, such as baseline TKV growth rate and Uosm, may be more relevant for predicting treatment efficacy than demographic characteristics. Thank you for this helpful comment. This important point is now made in the discussion, see discussion, paragraph 6

  1. Discussion Section (Lines 262-265): The authors mention that urinary osmolality (Uosm) may predict the efficacy of tolvaptan, citing a post-hoc analysis of the TEMPO 3:4 study, which correlated reductions in Uosm with a slower decline in kidney function.
    Suggestion: The authors correctly discuss the relevance of Uosm as a predictive biomarker. However, they could have explicitly linked this finding to Table S2, where Baseline Spot Uosm was identified as a significant predictor of response to tolvaptan (p = 0.03) in the logistic regression analysis. Additionally, the confidence interval for this effect (-0.014, -0.001) suggests that the relationship between this marker and treatment response still needs further investigation in larger studies. Thank you for this helpful comment. We now provide a reference to both Supplemental Table S2 and the main Table 2 for showing that baseline spot Uosm was identified as a significant predictor of tolvaptan response (p = 0.03) in the logistic regression analysis, page 8, line 211. We also include a recommendation for further investigation given the wide confidence interval for this effect, see discussion, paragraph 3.